# Tumor Grade and Mitotic Count Are Prognostic for Dogs with Cutaneous Mast Cell Tumors Treated with Surgery and Adjuvant or Neoadjuvant Vinblastine Chemotherapy

**DOI:** 10.3390/vetsci11080363

**Published:** 2024-08-10

**Authors:** Kristina Anderson, MacKenzie Pellin, Elizabeth Snyder, Dawn Clarke

**Affiliations:** 1Blue Pearl Pet Hospital, Southfield, MI 48034, USA; kristina.anderson@bluepearlvet.com; 2Department of Medical Sciences, Veterinary Medical Teaching Hospital, University of Wisconsin-Madison, Madison, WI 53706, USA; 3Blue Pearl Pet Hospital, Spring, TX 77388, USA; elizabeth.snyder@bluepearlvet.com; 4Department of Small Animal Medicine and Surgery, University of Georgia College of Veterinary Medicine, Athens, GA 30602, USA; dmclarke@uga.edu

**Keywords:** canine cutaneous mast cell tumor, vinblastine, neoadjuvant, grade

## Abstract

**Simple Summary:**

Canine cutaneous mast cell tumors have variable rates of recurrence and metastasis. We sought to evaluate how various prognostic factors affect survival, recurrence, and metastasis in a retrospective study of 90 dogs with cutaneous mast cell tumors treated with surgery and vinblastine chemotherapy. Factors evaluated included age, breed, grade, margins, local tumor control, mitotic count, lymph node metastasis, response to vinblastine chemotherapy in the gross disease setting, and timing of vinblastine in relation to surgery. Eighteen dogs received neoadjuvant vinblastine, and no dogs progressed locally before surgery. The use of neoadjuvant vinblastine was associated with a higher chance of local recurrence (*p* = 0.03) but not survival. Significantly shorter survival times were found for high-grade tumors (*p* < 0.001), grade 3 tumors (*p* < 0.001), and tumors with a mitotic count of >5 (*p* < 0.001). Dogs with grade 2/low grade cutaneous mast cell tumors lived longer than those with grade 2/high grade tumors (*p* < 0.001). Both grading systems and mitotic count were prognostic for survival in this population of dogs, supporting the need for standard reporting of histopathologic findings. Neoadjuvant chemotherapy can be effective in downsizing canine cutaneous mast cell tumors but does not influence survival.

**Abstract:**

Objective: Canine cutaneous mast cell tumors (cMCTs) have variable rates of recurrence and metastasis. We evaluated how various prognostic factors affect survival, recurrence, and metastasis in dogs with cMCT who underwent surgery and vinblastine chemotherapy. Animals: 90 dogs with cMCT treated with surgery and vinblastine at a veterinary referral institution were included. Methods: Medical records were retrospectively reviewed. Prognostic factors were evaluated. Results: Most dogs (94%) had grade 2 or 3 cMCTs. Neoadjuvant vinblastine was used in 18 dogs, and none progressed locally before surgery. The use of neoadjuvant vinblastine was associated with a higher chance of local recurrence (*p* = 0.03) but not survival. Shorter survival times were found for tumors that were high-grade (*p* < 0.001), grade 3 (*p* < 0.001), or a MC of >5 (*p* < 0.001). Dogs with grade 2 tumors that were low-grade lived longer than those with high-grade tumors (*p* < 0.001). Histologic tumor-free margins and the ability to achieve local tumor control were not associated with outcome. Clinical Relevance: Both grading systems and MC were prognostic for survival in this population of dogs, supporting the need for the standard reporting of histopathologic findings. Neoadjuvant chemotherapy can be effective in downsizing cMCTs but does not influence survival. These findings are consistent with previous publications, showing the benefits of a more modern population of patients, surgical treatments, and histopathologic assessments.

## 1. Introduction

Cutaneous mast cell tumors (cMCTs) are the most common malignant skin cancers seen in dogs [1]. cMCTs have a range of behaviors, from benign solitary masses to diffuse systemic disease. The biologic behavior of an individual tumor is difficult to determine on initial presentation alone, although breed and location, such as mucocutaneous junction, can sometimes predict behavior [2,3]. Canine cMCT are graded based on histologic criteria, and the course of disease is best prognosticated by histopathologic grading [4,5]; however, both two-tier and three-tier grading systems have their limitations. High interobserver variability is one of the larger pitfalls of the three-tiered system, while the two-tiered system may lack sensitivity [4,5]. Dogs with high-grade tumors have a median survival time of 3.6 months when treated with surgical excision alone and are significantly more likely to die due to their cMCT compared to low-grade tumors [4]. Controlling both the local disease and systemic disease is an ongoing challenge.

While the mainstay of treatment is complete surgical excision, various adjuvant treatments, such as radiation and chemotherapy, have been described and implemented based on the risk of local recurrence and metastasis. For grade 1 or 2 tumors with complete margins and a lack of overt metastasis, surgery alone may be adequate [6]. For dogs with high-grade/grade 3 tumors or evidence of overt metastasis, systemic therapy is widely considered necessary, although this has also been challenged [7]. Multiple chemotherapeutic agents have been suggested for the treatment of cMCTs, often in combination or with prednisone [8,9,10]. Vinblastine is one of the most studied chemotherapy agents used for canine cMCTs [11,12,13,14,15,16,17]. 

The ability to decrease tumor size with neoadjuvant therapy may allow for better local control with surgery, particularly for tumors on the extremities. Neoadjuvant therapy using prednisone and/or vinblastine is reported most frequently. A 63% median tumor volume reduction and a 45% tumor diameter reduction were noted in dogs with cMCT receiving neoadjuvant prednisone [18,19]. Response rates to neoadjuvant vinblastine are variable (12–47%), and most studies are small [13,14]. In a 1999 study [14], 47% of dogs had a measurable tumor response to neoadjuvant vinblastine, whereas a 12–27% overall response rate was found in a study comparing neoadjuvant vinblastine doses nine years later [13]. In a recent study conducted by Ossowska et al., 44 dogs with mast cell tumors were treated with neoadjuvant vinblastine in order to optimize the surgical margins or enable surgery. Measurable response was documented in 40.9% of dogs [20]. Local recurrence was documented in 20.8% of dogs, which were followed for at least 365 days after surgery [20]. The influence of neoadjuvant vinblastine therapy on progression and survival has not been extensively evaluated.

Previous studies have evaluated prognostic factors for dogs with cMCTs, such as histologic grading, age, breed, molecular markers, and tumor location [15,21,22,23,24,25]. Within the histologic grading criteria are other prognostic factors, such as mitotic count. It is speculated whether the mitotic count is independently prognostic when the grade is already accounted for in dogs with cMCTs [26]. The disadvantage of many prognostic factors in cMCTs is that they require biopsy and/or general anesthesia to gain this information; therefore, evaluating other clinical factors is important to make the best treatment plan for these patients. Achieving adequate disease control via regional lymphadenectomy also seems to be a key player in improving patient outcomes but is not as strong of a predictor of outcome as the two-tiered grading system alone [27].

Our primary objective was to evaluate the prognostic factors that influenced progression-free interval (PFI) and overall survival time (OST) in a contemporary population of dogs with cMCTs that underwent surgery and received vinblastine in either a neoadjuvant or adjuvant setting on an intent-to-treat basis. As previous publications were performed prior to the widespread utilization of mitotic counts for prognosis in cCMTs, this was an area of primary interest in our study. A secondary objective was to evaluate the outcomes of dogs receiving neoadjuvant vinblastine.

## 2. Materials and Methods

Case selection: A medical record database search at the University of Wisconsin Veterinary Care Hospital (UWVC) from 2009 to 2017 identified dogs diagnosed with a cMCT and that had received vinblastine chemotherapy and surgery performed with curative intent, i.e., debulking surgeries or excisional biopsies were not included unless followed by more definitive therapy. The dogs included had a histologic diagnosis of cMCT, no life-threatening comorbidities prior to the treatment start date, and received at least one dose of vinblastine. Dogs that received hypo-fractionated or hyper-fractionated radiation therapy prior to, concurrently, or after treatment with vinblastine were allowed if radiation therapy was deemed necessary to achieve adequate local control (ALC), consistent with previous papers evaluating the use of vinblastine in dogs with cMCTs [14,15,16]. Dogs were included if they had new or recurrent cMCTs or multiple cMCTs. Follow-up visits with UWVC or with the primary care veterinarian were at the discretion of the clinician and owner. Due to the retrospective nature of this study, follow-up information was collected from referring veterinarian records, either available through the UWVC medical record system or by calling referring veterinarians. Dogs were allowed any treatment modality, including follow-up surgery, chemotherapy, and radiation therapy, for recurrent tumors. Dogs were excluded if they lacked histopathologic grading, received concurrent systemic chemotherapy (such as toceranib phosphate) along with vinblastine, had a non-cutaneous tumor, or lacked sufficient follow-up. Insufficient follow-up included a lack of any veterinary visits after chemotherapy, missing or purged veterinary records after chemotherapy, or the inability to release post-chemotherapy veterinary records to UWVC.

Medical record review: Patient signalment and tumor characteristics were abstracted from medical records. Tumor size was not evaluated in this study as measurements were not consistently reported for several cases that had surgery prior to referral. Clinical staging prior to treatment was at the discretion of the attending clinician, and the results were recorded. Dogs with easily accessible or palpably enlarged lymph nodes (LNs) consistently had lymph node sampling performed. Not all dogs had local lymph node (LN) sampling (for example if not identifiable or too small to successfully sample), and spleen or liver aspirates to assess metastatic disease were not routinely performed if there were no ultrasonographic concerns for metastases in those organs. The extent of post-therapy follow-up and re-staging intervals were at the discretion of the attending clinician(s). Progressive disease (local recurrence and metastatic disease) was confirmed through evaluation of the medical records.

Treatment: As previously stated, all of the included dogs underwent surgery with curative intent. Due to the retrospective nature of the study, some dogs had primary surgery performed at a referral hospital prior to presenting at UWVC. For dogs who underwent surgery at UWVC, a surgical prescription of 2–3 cm wide and one fascial plane deep for margins was typically utilized when allowed by anatomical barriers. 

Radiation therapy, when given as part of the primary treatment plan, was evaluated when needed to obtain complete margins (i.e., not as a rescue after VBL had failed). Radiation therapy was delivered either prior to surgery, concurrently with VBL, or immediately after surgery. Dogs received either hypo-fractionated or hyper-fractionated radiation therapy. Hypo-fractionated protocols were administered with a prescribed protocol of 4 weekly fractions at 6.5–8 Gy each. Hyper-fractionated protocols were administered with 18–20 fractions, given Monday through Friday, with a total dosage of 48 Gy. 

Vinblastine was administered in the neoadjuvant or adjuvant setting or both. The intended protocol was 2.6 mg/m^2^ vinblastine given intravenously once weekly for 6 consecutive weeks. Prednisone was given concurrently at an initial dose of 2 mg/kg, then tapered over the 6 weeks of the vinblastine protocol. The dose escalation of vinblastine was at the discretion of the clinician. When used in the neoadjuvant setting, the number of doses given prior to surgery was clinician-dependent. For dogs with gross disease receiving neoadjuvant vinblastine, the Response Evaluation Criteria in Solid Tumors (RECIST, VCOG) was retrospectively applied to compare the largest tumor diameters [28]. When it was unclear whether the mass met the RECIST criteria for remission or progressive disease due to lack of measurements, the dog was considered to have “stable disease” if no mention of complete remission or disease progression were noted. 

Histopathologic features: Grade (Patnaik and/or Kiupel) and margin characteristics were collected from the original biopsy report or pulled from the patient records when the report was not available. Kiupel (high/low) grading was not available for all patients as it was not consistently reported in histologic reports until after the publication by Kiupel et al. in 2011. Patnaik grade (1–3) and mitotic count (MC), reported as the number of mitotic figures per 10 high-powered fields (hpf), were available for all dogs. If a dog had multiple tumors at the time of presentation or treatment, the highest-grade tumor was followed. We used two different cut-offs to categorize the MC based on previous publications [25,28]. Tumors were grouped as ≥ 5 and <5 mitotic figures per 10 hpfs, as well as <1, 1–7, and >7 mitotic figures per 10 hpfs for the evaluation of mitotic count as a prognostic factor.

Margins were categorized for statistical analysis using a cut-off of 3 mm using the narrowest deep or lateral margin noted. The adequacy of local control was determined after the completion of local therapies (surgery, ±neoadjuvant vinblastine, ±radiation therapy). Adequate local control (ALC) was defined as those dogs in which the narrowest histopathologic margin was ≥3 mm or those with a margin <3 mm that received definitive radiation therapy. Adjuvant vinblastine treatment in the face of narrow (<3 mm) surgical margins was not considered ALC. All dogs with known metastatic LNs on cytology had lymphadenectomy to be considered to have adequate local control. Sentinel LNs were not consistently removed without clinical suspicion of metastatic disease.

Statistical analysis: Descriptive statistics were recorded for patient and tumor characteristics. Kolmogorov–Smirnov and Shapiro–Wilk tests for normality were used. For any variables with a non-normal distribution, non-parametric tests were used, and the results are reported as median ± 95% confidence interval (CI). Fisher’s exact tests were used to analyze 2 × 2 tables when one or more of the values was less than 5. Two-tailed, unpaired *t*-tests, Mann–Whitney, Fisher’s exact or chi-square tests were used to compare variables.

Dogs were censored from survival analysis if they were lost to follow-up or still alive at the end of the study. Overall survival time (OST) was recorded as the time of treatment start (either surgery or first VBL) to the date of death from any cause. The cause of death was unknown in many cases; therefore, unless another cause was known, death was assumed to be due to a mast cell tumor. The progression-free interval (PFI) was recorded as the time of treatment start (either surgery or first VBL) to cMCT progression (either local recurrence or metastasis). The development of de novo tumors (new tumor formation elsewhere on the body not regionally associated with the original tumor) after therapy was not considered progressive disease. The median OST and PFI were calculated using the Kaplan–Meier method.

Variables evaluated as predictors of progression or survival included age, body weight, Kiupel (when available) and Patnaik grade, MC, LN metastasis at diagnosis, margins, adequacy of local control, neoadjuvant chemotherapy, and local recurrence. Log-rank was used to assess prognostic factors for influence on PFI and OST. The results are presented as median (±95% CI). Cox proportional hazards analysis was used in univariate analysis, and the results are presented as hazards ratio (HR) ± 95% CI. Univariate prognostic variables with a *p* value < 0.1 were evaluated in multivariate analysis using Cox proportional hazards analysis using a step-wise approach. Due to correlations between mitotic count and both histologic grading schemes, these variables were not included in models together, but rather analyzed separately with other prognostic factors. A *p*-value of <0.05 was considered significant. Statistical analyses were performed using SPSS v.28.0.1.1, IBM, Armonk, NY, USA.

## 3. Results

### 3.1. Patient Characteristics and Staging 

Records from 136 dogs were available for review, and 90 dogs met the inclusion criteria. The remaining 46 dogs were excluded due to the following reasons: non-cutaneous MCT, surgery lacking curative intent, concurrent chemotherapy along with vinblastine, or lack of sufficient follow-up. At the time of presentation, 10 dogs (11%) had recurrent cMCTs and 25 dogs (28%) had multiple cMCTs recorded. Various breeds were represented, with Labrador retrievers and mixed breeds being the most common (Table 1). 

Staging was performed at the discretion of the clinician (Table 1). The CBC was documented as normal in the records of 69 dogs (79%), and leukocytosis was documented in 7 dogs (8%). Pre-treatment serum chemistries were clinically unremarkable in 47 dogs (58%), with the most common abnormalities being elevation in liver values in 22 dogs (28%). Lymph node sampling (fine needle aspirate cytology and/or histopathology) was performed in 64 dogs (71%), and LN metastasis at the time of vinblastine treatment was confirmed in 30 dogs (47%). No dogs had histologic LN grading. Abdominal ultrasound was performed in 82 (91%) dogs and liver, and/or splenic aspirates were recorded in 19 (21%) dogs. One dog (5%) had cytologic confirmation of metastasis to the liver and spleen. Of the dogs that received an abdominal ultrasound, 41 (50%) were considered normal, while mild heterogenous hepatopathies, mildly enlarged medial iliac lymph nodes, benign-appearing splenic nodules, and mild degenerative renal disease were noted in the remaining dogs. 

### 3.2. Histologic Characteristics

All dogs had a Patnaik grading of their cMCT, and the majority were grade 2 (61%) (Table 1). Only 5 dogs (6%) had grade 1 tumors, and 30 dogs (33%) had grade 3 tumors. A Kiupel grade was not recorded in 17 dogs (19%). Of the dogs that had their Kiupel grade recorded, 35 dogs (49%) had low-grade tumors, and 37 dogs (51%) had high-grade tumors. The median mitotic count (MC) per 10 high-powered fields (hpfs) was 2 (range 0–38). 

We sought to determine if there was an association between histopathologic grade or mitotic count and metastasis (Table 2). The presence of LN metastasis in the dogs who underwent LN sampling was not associated with a higher Patnaik grade (*p* = 0.78) or high Kiupel grade (*p* = 0.27). Similarly, the presence of lymph node metastasis at the time of diagnosis was not associated with mitotic count category for either mitotic count cut-off (*p* = 0.72 and *p* = 0.94, respectively). Developing metastasis (local or distant) after vinblastine treatment was not associated with either grading system (*p* = 0.78 for Patnaik and *p* = 0.25 for Kiupel) but developing metastases after treatment occurred more often in dogs with tumors in the highest mitotic count category using either mitotic count cut off (*p* = 0.04 and *p* = 0.03, respectively) (Table 2). 

### 3.3. Local Tumor Control

Histopathology reports and patient records were evaluated to retrospectively determine the completeness of local control. At least one histologic margin measurement was available for all dogs. The median lateral and deep histologic margins were 1.5 mm and 1 mm (range 0–20 mm), respectively. To determine the influence of surgical margins on prognosis, margin cut-offs of 3 mm and 5 mm were both evaluated. Associations with other variables and survival were similar using both cut-offs, so the narrower 3 mm cut-off was used in subsequent analyses. There was a higher likelihood of achieving a margin ≥3 mm in grade 1 tumors compared to grade 2 or 3 tumors (100% for grade 1 vs. 34% for grades 2 and 3; *p* = 0.006), but complete margins were not associated with Kiupel grading (42% for low grade vs. 35% for high grade; *p* = 0.57). Dogs with LN metastasis at diagnosis more often had incomplete margins (*p* = 0.04). 

Adequate local control (ALC) after therapy (accounting for neoadjuvant vinblastine, surgery, and radiation therapy) was achieved in 37 dogs (41%). There were only two dogs (2%) that had gross disease after treatment. One dog had liver and spleen metastases, and another dog had gross disease remaining after attempted excision, which progressed after one dose of post-operative radiation therapy. Due to the small number of dogs in the category of gross disease, these two dogs were excluded from the analysis of ALC. All dogs with Patnaik grade 1 tumors had adequate local control compared to only 39% of those with grade 2 or 3 (*p* = 0.007). Kiupel grade, use of neoadjuvant vinblastine, and LN metastasis at diagnosis did not differ in dogs with ALC compared to those that were inadequate (*p* = 0.48, *p* = 0.07, and *p* = 0.59, respectively).

### 3.4. Treatment

Vinblastine was started post-operatively in 72 dogs, a median of 21 days after surgery (range 8–115 days) (Table 3). Only 11 dogs (12%) started vinblastine >31 days after surgery. The mean starting vinblastine dose was 2.5 mg/m^2^ (range 1.6–2.7 mg/m^2^). Dose escalation was at the discretion of the clinician, and the maximally tolerated dose ranged from 1.4 to 3.8 mg/m^2^. Nineteen dogs (21%) received fewer than the six planned vinblastine doses and five dogs (6%) received more than six doses (range 7–13). A total of 487 doses of VBL were administered, and adverse events (AEs) were noted in 43 doses (9%), including nausea, fever, lethargy, or neutropenia. A dose reduction was made in 33 dogs (37%) due to a significant adverse event (grade 3 or 4 toxicity) or grade 2 neutropenia resulting in dose delay. The first VBL dose was the dose most often associated with an adverse event. Neutropenic events were noted in 37 dogs (41%); 10 were grade 4, 11 were grade 3, 13 were grade 2, and 3 were grade 1.

Eighteen dogs (20%) were treated with neoadjuvant vinblastine with the intention of decreasing tumor size before surgery (Table 3). The median number of neoadjuvant doses of VBL was 3. Patient and tumor characteristics between those that received neoadjuvant versus only adjuvant vinblastine were similar, except for the fact that grade 1 tumors were overrepresented in the neoadjuvant group (*p* = 0.03). Grade 2–4 neutropenia occurred in 8/18 (44%) dogs. When comparing overall toxicity between the adjuvant and neoadjuvant groups, there was no significant difference in the development of neutropenia or a significant adverse event (*p* = 0.21 and *p* = 0.58, respectively). Exact tumor measurements after vinblastine were not recorded in the records of five dogs, but written descriptions of the mass were available to confirm that progressive disease did not occur. For those with available measurements, 77% of dogs had an objective response according to RECIST criteria [27] (Table 3). Twelve dogs received additional vinblastine after surgery.

Prednisone was administered concurrently in 59 dogs (82%) receiving adjuvant vinblastine and 17 dogs (94%) receiving neoadjuvant vinblastine. Prednisone was administered at a mean dose of 1.1 mg/kg (0.5–2.3 mg/kg) for all dogs. Other concomitant medications included diphenhydramine in 56 dogs (62%) and a gastrointestinal acid suppressant medication (famotidine or omeprazole) in 62 dogs (69%). Dosages for these supportive medications were not available for all patients.

Radiation therapy (RT) was administered after surgery in 11 dogs (12%) (Table 1). Seven dogs who received adjuvant vinblastine only had radiation therapy: three dogs completed RT prior to starting vinblastine, two dogs received radiation therapy concurrently with vinblastine, one dog began RT after completing vinblastine, and one dog received radiation therapy after vinblastine failure. Four dogs who received neoadjuvant vinblastine received RT; none had RT prior to surgery. Two dogs received RT post-operatively, and two received it in the rescue setting. There were no differences in patient and tumor characteristics between dogs who did or did not receive radiation therapy.

### 3.5. Outcome

The median survival time for all dogs was 831 (95% CI = 415–1246) days, with a range of 20 to 3854 days. When evaluating the 64 dogs that received the intended protocol (≥6 vinblastine doses), median OST was 1111 (95% CI = 59–1462) days. At the time of analysis, 63 dogs (70%) had died. The median follow-up time was 1777 (range, 614–3691) days for the 13 dogs (14%) alive at the time of analysis. For the 14 dogs lost to follow-up, their median follow-up time was 133 (range, 32–1545) days. A necropsy report was only available for three dogs (3%), and all had progressive mast cell disease at the time of death. A cause of death was unknown in 28 dogs (44%), 32 dogs (51%) died due to their cMCT, and 3 dogs (5%) died of other causes, including other cancers (2) and heart disease (1). One-, two- and three-year survival rates for all dogs were 63%, 47%, and 32%, respectively. Of the 26 dogs that died within 1 year, 20 dogs (77%) died of their cMCT, whereas only 8 of 57 dogs (14%) that died after 1 year died of their cMCT (*p* < 0.001). 

The median progression-free interval (PFI) was 920 days (range 20–3854) days. cMCT progression occurred in 42 dogs (47%), and of those, 31 dogs (34%) had local recurrence, and 25 dogs (28%) developed metastatic disease (lymph node and/or spleen/liver). De novo tumor formation occurred in 17 dogs (19%) after treatment completion but was not considered progressive disease in survival analysis. Rescue chemotherapy (including toceranib phosphate, lomustine, chlorambucil, etc.) was used in 26 dogs (29%) after surgery/vinblastine treatment for disease progression, ranging from days to years after their treatment. 

Factors associated with local recurrence and the development of metastatic disease after treatment were evaluated. Local recurrence was more frequent in dogs with Kiupel high-grade tumors compared to low-grade tumors (49% vs. 25%; *p* = 0.04). Patnaik grade and mitotic count were not associated with local recurrence. Margins of <3 mm did not significantly increase the chance of local recurrence (39% recurrence with margins <3 mm vs. 27% with margins ≥3 mm; *p* = 0.22). Even when factoring in post-op radiation therapy, lack of adequate local control did not significantly increase the chance of local recurrence (27% recurrence with ALC vs. 37%; *p* = 0.32). Dogs that developed local recurrence had a shorter survival than those without recurrence (304 vs. 1290 days, *p* < 0.001). Tumor grade using either the Patnaik or Kiupel grading systems was not associated with the development of metastasis after treatment (*p* = 0.10 and *p* = 0.22, respectively). However, the mitotic count was significantly associated with the development of metastasis after treatment. Forty percent of dogs with a MC ≥5 developed metastatic disease compared to only 20% of those with a MC <5 (*p* = 0.04). Dogs with local recurrence were also more likely to develop metastasis after treatment compared to those who did not recur (56% vs. 44%; *p* = 0.008).

### 3.6. Survival Analysis

#### 3.6.1. Neoadjuvant Vinblastine

There was no difference in OST or PFI in dogs that received neoadjuvant VBL vs. adjuvant VBL (Table 4). Other clinical and treatment parameters were similar between those that received neoadjuvant or adjuvant VBL other than there were more dogs with grade 1 tumors among the neoadjuvant dogs (*p* = 0.03). Local cMCT recurrence was seen in 56% of dogs that received neoadjuvant vinblastine compared to 29% in dogs without neoadjuvant therapy (*p* = 0.04). 

#### 3.6.2. Lymph Node Metastasis

LN metastasis at diagnosis was not associated with Kiupel grade, Patnaik grade, or mitotic count (*p* = 0.27, *p* = 0.78, *p* = 0.72, respectively). Dogs that had LN metastasis at diagnosis had a shorter PFI (173 days) compared to dogs that did not (PFI not reached) (*p* = 0.03; Table 3). Dogs with LN metastasis had a higher likelihood of developing local recurrence compared to those without LN metastasis (60% vs. 20%; *p* = 0.001). However, LN metastasis at diagnosis did not predict the development of further metastasis after treatment (*p* = 0.62). Interestingly, the presence of LN metastasis at diagnosis was not associated with survival (*p* = 0.35) overall. However, when evaluating only dogs with Kiupel high-grade or Patnaik grade 3 tumors, having lymph node metastasis at diagnosis was significantly associated with shorter survival (*p* = 0.002 and *p* = 0.005, respectively).

#### 3.6.3. Histopathologic Characteristics

Patnaik and Kiupel grading were both strongly associated with survival (*p* < 0.001 and *p* < 0.001, respectively). Similarly, dogs with grade 3 or high-grade tumors had a shorter PFI (*p* = 0.04 and *p* = 0.009, respectively) (Figure 1, Table 4). To determine whether there is a benefit in using both MCT grading systems, we evaluated only dogs with Patnaik grade 2 tumors separately. For dogs with grade 2 tumors, the OST was significantly longer when tumors were classified as Kiupel low-grade (1798 days) compared to Kiupel high-grade (466 days) (*p* < 0.001) (Figure 2). PFI was also higher in dogs with grade 2 tumors that were Kiupel low-grade compared to those that were Kiupel high-grade; however, this difference was not significant (*p* = 0.07). Higher MC was also associated with shorter OST and PFI when using either >5 or >7 as cut-offs (Figure 1, Table 4).

#### 3.6.4. Multivariable Analysis

Multivariable models for PFI and OST, including age, LN metastasis, local tumor control, and either Kiupel grade, Patnaik grade, or mitotic count, were evaluated. Adequate local control was not associated with PFI when controlling for other factors in any of the models (Table 5). Dogs with LN metastases were more likely to experience disease progression in multivariable models of Kiupel, Patnaik, or mitotic count [HR 3.5 (1.5–8.1) *p* = 0.003; HR 2.5 (1.2–5.2) *p* = 0.013, and HR 2.8 (1.2–6.1) *p* = 0.009, respectively] (Table 5). Age at diagnosis was independently associated with risk of death when controlling for LN metastasis, local control, and histopathologic grade or mitotic count. Kiupel and Patnaik histopathologic grade and high MC were associated with a higher risk of death independent of age, LN metastasis at diagnosis, and local tumor control [HR 3.2 (1.4–7.2) *p* = 0.004; HR 2.9 (1.5–5.7) *p* = 0.002, and HR 2.8 (1.4–5.5) *p* = 0.003, respectively] (Table 5). The ability to achieve local tumor control and the presence of LN metastases at diagnosis were not significantly associated with the risk of death in multivariable analysis.

## 4. Discussion

This retrospective case series evaluated previously established cMCT prognostic factors in a contemporary population of dogs undergoing surgery and vinblastine chemotherapy. The 6-week course of vinblastine chemotherapy used in this study was found to be well tolerated overall. Thirty-seven percent of patients did require a dose reduction; however, this included patients with a grade 2 neutropenia, given the weekly nature of the vinblastine protocol. Our patient population was similar to those that have been previously published. Labrador retrievers were the most common breed amongst our patient population and have previously been reported to be predisposed to mast cell tumor development [29]. Histopathologic grading and MC remain strong prognostic indicators for dogs with cMCTs. LN metastasis at diagnosis and adequate local control were not prognostic for survival in dogs treated with surgery and VBL. In a subset of these patients, we found that neoadjuvant vinblastine is effective for downsizing cMCT and does not influence PFI or OST; however, local recurrence was more common. 

Histologic grading continues to be one of the strongest predictors of PFI and OST in cMCT. A 2020 consensus statement on the grading of MCTs recommended reporting both the Kiupel and Patnaik grading systems, finding the Kiupel system to be more specific and the Patnaik system to be more sensitive [6]. By combining both systems, we showed that dogs with grade 2/low-grade cMCT had a longer OST and PFI than those with grade 2/high-grade cMCT, which helps predict the behavior of grade 2 tumors. However, due to our inclusion criteria of only dogs who received VBL, we had few grade 1 and low-grade cMCTs, making generalized conclusions on the grading systems difficult [11,13,14,15,16]. Our study supports the need to further evaluate using both grading systems to help prognosticate canine cMCTs in a more diverse population. 

Meuten et al. advocated for the adoption of the term mitotic count instead of mitotic index. Mitotic count is defined as the number of cells in mitosis in a standardized area (2.37 mm^2^) compared to mitotic index, which is the number of cells undergoing mitosis divided by the number of cells not undergoing mitosis [30,31]. As suggested by Meuten et al., standardization of reporting is not yet commonly practiced. Mitotic count, although a more objective measurement to prognosticate cMCTs in dogs compared to grading, has not been extensively evaluated in previous studies [13,14]. Mitotic count is considered in both grading systems [4,5]. Romansik et al. found that mitotic count was a strong predictor of survival in dogs with cMCTs, showing that dogs with a MC of 0–5 survived 70 months compared to only 2 months for those with a MC >5 [26]. In our population, dogs with a MC 0–5 survived 49 months compared to 10 months for dogs with a MC >5. Using a more analytical approach to MC groupings, Elston et al. found that MC cut-offs of <1, 1–7, and >7 were more appropriate [29]. In our study, both groups were strongly associated with PFI and OST, and only small differences were noted between the different groupings, most likely due to similar subgroups. Future studies with a larger, more diverse patient population could help distinguish which MC cut-off best prognosticates cMCTs. 

The six-dose weekly vinblastine protocol used at the UWVC for dogs with cMCT differs from the widely accepted eight-dose protocol [10,11,12,13,14,15,16,32,33,34,35]. There are limited studies using a six-dose weekly 2.6 mg/m^2^ VBL protocol for dogs with cMCTs, and its influence on survival and metastatic rate when compared to the more widely used eight-dose protocol is unknown [13,14,15,16]. In our study, the 64 dogs who received the six-dose protocol only in the adjuvant setting had an OST of 1111 (759–1462) days and 42% 3-year survival rate. In previous studies using the eight-dose protocol, MST for all dogs was not reached, and 3-year survival was 65% [13,14]. A case–controlled prospective study comparing the efficacy of the eight-dose protocol and this six-dose protocol is warranted.

Neoadjuvant therapies are prescribed to help downsize cMCTs to facilitate surgical removal. A previous retrospective study evaluating the use of neoadjuvant prednisone for dogs with cMCTs showed an overall objective response of 70%, and the median maximum diameter reduction was 45% [19]. Similarly, a more recent study found an overall tumor volume reduction of 64% [18]. This study is only the second to look at neoadjuvant vinblastine. In a recent study by Ossowska et al., 44 dogs with mast cell tumors were treated with neoadjuvant vinblastine and a measurable response was documented in 40.9% of dogs [20]. Several studies have evaluated the treatment of measurable cMCT with vinblastine with remission rates of 12–27% for a response duration of 77–154 days [13,14]. In a study comparing VBL to toceranib phosphate in macroscopic cMCT, dogs who received vinblastine had an overall response rate of 30% [36]. Neoadjuvant vinblastine can be useful in dogs with the aim of achieving surgical control when surgery is not otherwise deemed possible. There is consensus that dogs with no measurable disease live longer and have a longer PFI compared to dogs with gross disease [8,10,13,14,15,35,36]. In our study, the use of neoadjuvant vinblastine induced remission in 55%, and no dogs progressed from baseline while receiving therapy. However, because we considered the dogs to have stable disease when tumor measurements were unknown, RECIST-defined remission could have been underestimated. We also found that dogs receiving neoadjuvant vinblastine were significantly more likely to experience local recurrence despite having adequate local control. This sparks the question of whether the definition of adequate local control differs for dogs that receive neoadjuvant vinblastine. Ultimately, the impact of neoadjuvant vinblastine on histologic surgical margins needs to be further evaluated. 

Data on the long-term outcome of dogs treated with neoadjuvant vinblastine is quite limited. In a recent study conducted by Ossowska et al., local recurrence was documented in five (20.8%) dogs who were followed for at least 365 days after surgery, and no data were evaluated in terms of survival or development of metastasis [20]. In our study, neoadjuvant vinblastine did not alter PFI or OST. However, variability in post-surgical vinblastine dosing could have altered outcomes. In addition, there were more grade 1 and low-grade tumors in the neoadjuvant group, which could have confounded the outcome. A further limitation is that many dogs receiving neoadjuvant vinblastine also went on to receive adjuvant vinblastine as well, making the influence of the neoadjuvant vinblastine difficult to assess. Future research is called for to further elucidate the impact of neoadjuvant vinblastine chemotherapy on cMCTs. It is also important to consider that neoadjuvant vinblastine could alter two important prognostic factors: histologic grading or MC. Two recent studies showed that neoadjuvant prednisone did not alter MCT grade; however, this should be evaluated in a prospective study of neoadjuvant vinblastine due to the different mechanisms of action [18,37]. The use of prednisone may have also influenced the results of the study; however, it is standard practice at our institution to combine prednisone with vinblastine, making it difficult to assess the individual impact of each medication.

While there are many studies evaluating the most ideal surgical margins to prevent local recurrence for cMCTs, there are limited studies prospectively evaluating the most ideal histologic margin [38,39,40]. The histologic margin needed to prevent cMCT recurrence is unknown and may not exist, as local recurrence can be multifactorial. In a 2013 study, Donnelly et al. were unable to identify a histological tumor free margin width that would prevent local recurrence [41]. High-grade tumors that recurred and those that did not both had a median histologic margin of 4 mm. In our study, when assessing histologic margins, incomplete or narrow margins (<3 mm) were not prognostic for local recurrence, PFI, or MST. 

Lymph node metastasis at cMCT diagnosis is a debated prognostic factor [7,15,27,42,43,44,45]. In our study, the presence of LN metastasis at diagnosis was prognostic for PFI but not OST, and this was still evident in multivariable analysis. This is likely attributed to the use of chemotherapy and surgery in our population. Guerra et al. found that histologic tumor grade was a stronger predictor of outcome compared to nodal status, and our results corroborate that finding [27]. The inability to document and sample the sentinel lymph node may also contribute to this finding. With improved technologies aimed at appropriate lymph node mapping in dogs with cMCTs, it is possible that more significant associations may be found [46]. Only one dog with a grade 1 tumor had LN metastasis at diagnosis in our study. Half of the dogs with LN metastases had high-grade tumors. One of the biggest limitations of our study was the lack of standardized staging; only 70% of the dogs had documented LN sampling at the time of diagnosis. When only considering dogs with aggressive tumors (high-grade or grade 3), LN metastasis was found to be associated with a poorer OST and PFI. Distant metastases may also be underreported in our study, given the lack of standardized staging at initial diagnosis and during follow-up. At the University of Wisconsin Veterinary Care Hospital, spleen and liver aspirates are not routinely performed in every case and are instead dependent upon imaging findings; for some patients with ultrasonographic changes to these organs, sampling was not performed based upon the ranking of benign processes being more highly or due to clients declining aspirates. Other ultrasonographic abnormalities were also documented in organs unlikely to contain mast cell tumor metastases (such as degenerative kidney changes). While we cannot rule out the higher incidence of distant metastases at treatment initiation due to these limitations, no patients with these mild changes clinically declined, suggesting that the changes were not secondary to mast cell disease.

In addition to the limitations listed above, the retrospective nature of this study, lack of follow-up, and variable treatment protocols after progression made assessing prognostic factors challenging. Our study population is biased to only portray the prognosis of dogs receiving both surgery and vinblastine and does not reflect the prognosis of all dogs with cMCTs. The inability to have a single pathologist review all slides likely resulted in some inherent subjectivity in terms of grade, margins, and MC. In addition, not all dogs had the Kiupel grading scheme applied. Recheck frequency was not standardized, and the recording of de novo cMCTs, metastasis, and recurrent cMCTs in referral records was not always clearly documented. A larger number of patients and more consistent follow-up would allow for more concrete data to prove the associations between prognostic factors in multivariable analysis. The number of dogs treated with neoadjuvant vinblastine was also relatively small, making conclusions on its efficacy and influence on outcome difficult. The population of dogs that received neoadjuvant vinblastine was likely biased, as dogs with larger tumors are clinically more likely to receive neoadjuvant chemotherapy to make surgery feasible, and owners willing to pursue neoadjuvant therapy may be more financially and emotionally prepared to consent to the most aggressive treatments.

## 5. Conclusions

Histopathologic grading and MC remain strong prognostic factors for outcomes in dogs with cMCTs. Neoadjuvant VBL can effectively downsize cMCT, although those patients experienced higher rates of local recurrence and did not experience a survival benefit. LN metastasis at diagnosis and completeness of surgical margins were not prognostic for survival in all dogs with cMCT treated with surgery and VBL chemotherapy. Future studies with a larger, more diverse patient population could help determine the best prognostic factors for cMCT and the efficacy and influence of neoadjuvant vinblastine on prognosis.

## Figures and Tables

**Figure 1 vetsci-11-00363-f001:**
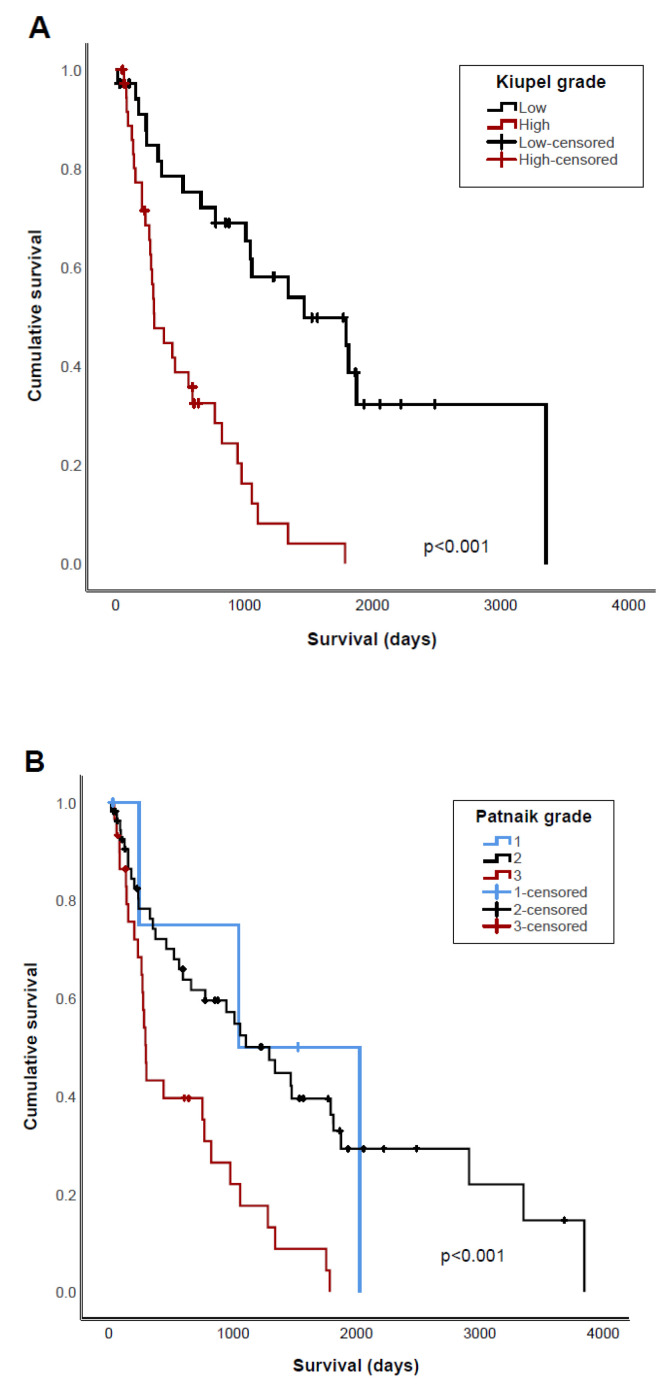
Kaplan–Meier survival curves of overall survival for dogs with cMCT comparing (**A**) high or low Kiupel grade; (**B**) Patnaik grade 1, 2, or 3; (**C**) mitotic count ≥5 or <5 per 10 high-powered fields; and (**D**) mitotic count <1, 1–7, or >7 per 10 high-powered fields.

**Figure 2 vetsci-11-00363-f002:**
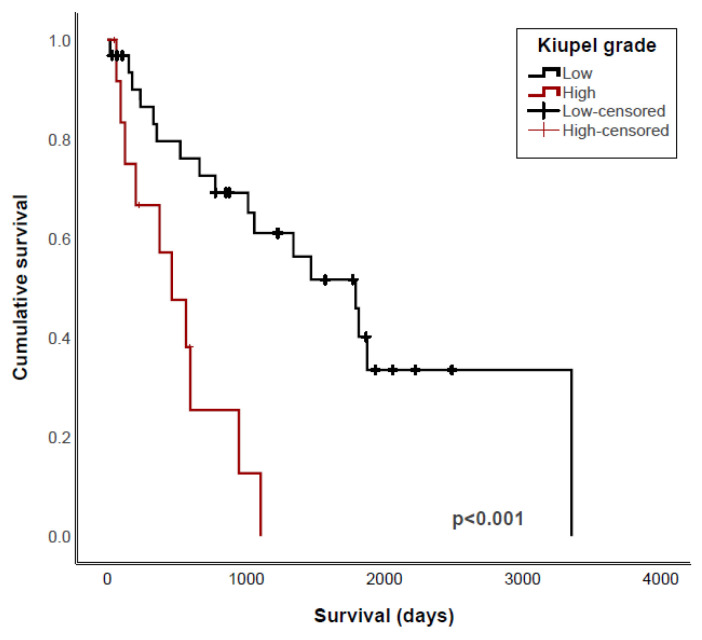
Kaplan–Meier survival curves of overall survival for 55 dogs with Patnaik grade 2 cutaneous mast cell tumors comparing overall survival time between Kiupel low- and high-grade tumors.

**Table 1 vetsci-11-00363-t001:** Patient population and tumor characteristics of 90 dogs diagnosed with cMCT treated with surgery and vinblastine chemotherapy.

Age (mean, range)	7.8 (3–16) years
Weight (mean, range)	26.3 (2.2–52.7) kg
Gender	
Spayed female	48 (53.3%)
Intact female	3 (3.3%)
Neutered male	36 (40%)
Intact male	3 (3.3%)
Breed	
Labrador retriever	26 (29%)
Mixed breed	25 (28%)
Golden retriever	6 (7%)
Boxer	5 (6%)
Boston terrier	4 (4%)
Other pure breed	24 (26%)
MCT location	
Limb	27 (30%)
Trunk	22 (25%)
Head/neck	19 (21%)
Others	12 (13%)
Multiple	10 (11%)
Initial diagnostics	
Chest radiographs	73 (82%)
Abdominal ultrasound	82 (91%)
Regional LN sampling	62 (69%)
Number of VBL doses (mean, range)	6 (1–13)
VBL starting dose (median, range)	2.5 (1.6–2.7) mg/m^2^
VBL SAE (# dogs, %)	33 (37%)
Radiation therapy	
Palliative (6.5–8 Gy fractions ( 4 weekly)	4 (4%)
Definitive (48 Gy in 18–20 fractions M–F)	7 (6%)
Concomitant medications	
Prednisone	76 (84%)
Diphenhydramine	56 (62%)
GI protectant	62 (69%)

Abbreviations: LN = lymph node, RT = radiation therapy, SAE = serious adverse event, VBL = vinblastine, # = number.

**Table 2 vetsci-11-00363-t002:** Comparisons of grade and mitotic count to outcomes in dogs with cMCT.

	Patnaik Grade				Kiupel Grade			Mitotic Count (per 10 hpf)			Mitotic Count (per 10 hpf)			
	1	2	3	*p*-Value	Low	High	*p*-Value	<5	>5	*p*-Value	<1	1–7	>7	*p*-Value
**Total number of dogs**	5 (6%)	55 (61%)	30 (33%)		36 (49%)	37 (51%)		55 (61%)	35 (39%)		17 (19%)	45 (50%)	28 (31%)	
**Number with LN sampling**	3 (60%)	38 (69%)	23 (77%)	0.65	26 (72%)	26 (70%)	0.85	42 (76%)	22 (63%)	0.17	14 (82%)	33 (73%)	17 (61%)	0.27
**Lymph node metastasis**	1 (33%)	19 (50%)	10 (44%)	0.78	10 (39%)	14 (54%)	0.27	19 (45%)	11 (50%)	0.72	6 (43%)	16 (49%)	8 (47%)	0.94
**Distant metastasis**	0	0	1 (1%)	–	0	1 (1%)	–	0	1 (1%)	–	0	0	1 (1%)	–
**Local recurrence**	2 (50%)	16 (32%)	13 (48%)	0.34	10 (29%)	17 (57%)	**0.03**	15 (29%)	16 (55%)	**0.02**	4 (24%)	16 (39%)	11 (48%)	0.29
**MCT progressive disease**	3 (60%)	34 (62%)	11 (37%)	0.08	23 (64%)	15 (40%)	0.05	35 (64%)	13 (37%)	**0.01**	12 (71%)	25 (56%)	11 (39%)	0.11
**Metastasis after treatment**	0	13 (24%)	12 (40%)	0.10	8 (22%)	13 (35%)	0.22	11 (20%)	14 (40%)	**0.04**	3 (18%)	9 (20%)	13 (46%)	**0.03**
**1-year survival**	3 (60%)	41 (75%)	13 (43%)	**0.02**	29 (81%)	18 (49%)	**0.004**	41 (75%)	16 (46%)	**0.006**	16 (94%)	31 (69%)	10 (36%)	**<0.001**
**2-year survival**	3 (60%)	30 (55%)	9 (30%)	**0.08**	13 (36%)	28 (76%)	**<0.001**	34 (62%)	8 (23%)	**<0.001**	14 (82%)	23 (51%)	5 (18%)	**<0.001**
**3-year survival**	2 (40%)	23 (42%)	4 (13%)	**0.03**	17 (47%)	3 (8%)	**<0.001**	26 (47%)	3 (9%)	**<0.001**	11 (65%)	16 (36%)	2 (7%)	**<0.001**

Abbreviations: LN = lymph node; hpf = high power field; – = not applicable; Bold results = statistical significance.

**Table 3 vetsci-11-00363-t003:** Differences in tumor and patient characteristics of dogs who received neoadjuvant vinblastine compared to those who did not.

	Neoadjuvant VBL 18 Dogs	Adjuvant VBL 72 Dogs	*p*-Value
No. neoadjuvant VBL doses	6 (2–6)	NA	
No. VBL doses	6 (2–13)	6 (2–6)	0.09
Response to VBL		NA	
Complete	1 (8%)		
Partial	9 (69%)		
Stable disease	3 (23%)		
Progressive disease	0 (0%)		
Not assessed	5		
Neutropenia grade			0.24
1	1 (12%)	2 (7%)	
2	5 (63%)	8 (28%)	
3	1 (12%)	10 (34%)	
4	1 (12%)	9 (31%)	
Serious adverse event			0.38
No	13 (72%)	44 (61%)	
Yes	5 (28%)	28 (39%)	
Patnaik grade			**0.03**
1	3 (17%)	2 (3%)	
2	1 (67%)	43 (60%)	
3	3 (16%)	27 (37%)	
Kiupel grade			0.10
Low	11 (69%)	25 (44%)	
High	5 (31%)	32 (56%)	
Mitotic count			0.28
<5 per 10 hpf	13 (72%)	42 (58%)	
>5 per 10 hpf	5 (28%)	30 (42%)	
Mitotic count			0.65
<1 per 10 hpf	4 (22%)	13 (18%)	
1–7 per 10 hpf	10 (56%)	35 (49%)	
>7 per 10 hpf	4 (22%)	24 (33%)	
LN metastasis			0.95
No	7 (54%)	27 (53%)	
Yes	6 (46%)	24 (47%)	
Histologic margins			0.43
<3 mm	9 (53%)	45 (63%)	
≥3 mm	8 (47%)	26 (37%)	
Adequate local control			0.07
No	7 (39%)	44 (63%)	
Yes	11 (61%)	26 (37%)	
Local recurrence			**0.03**
No	8 (44%)	51 (71%)	
Yes	10 (56%)	21 (29%)	
Metastasis			0.24
No	11 (61%)	54 (75%)	
Yes	7 (39%)	18 (25%)	
Prednisone			0.19
No	1 (6%)	13 (18%)	
Yes	17 (94%)	59 (82%)	

Abbreviations: hpf = high powered field, No. = number of; LN = lymph node VBL = vinblastine; Bold results = statistical significance, NA = not applicable.

**Table 4 vetsci-11-00363-t004:** Results of univariate analysis of potential prognostic factors for progression-free interval and overall survival in 90 dogs with cMCT treated with surgery and vinblastine.

	No. Dogs	PFI (±95% CI) (days)	*p*-Value	OST (±95% CI) (days)	*p*-Value
Patnaik grade			**0.04**		**<0.001**
1	5	NR		1051 (442–2240)	
2	55	NR		1300 (1168–2006)	
3	30	251 (161–341)		300 (390–802)	
Kiupel grade			**0.009**		**<0.001**
Low	36	NR		1475 (1265–2230)	
High	37	211 (47–375)		304 (382–697)	
Histo margins			0.46		0.76
<3 mm	54	364 (NR)		831 (210–1452)	
≥3 mm	34	1070 (0–2431)		1019 (509–1529)	
Adequate local control			0.35		0.72
No	51	364 (NR)		782 (229–1335)	
Yes	37	1070 (NR)		984 (382–1585)	
Mitotic count			**0.002**		**<0.001**
≤5 per 10 hpf	55	NR		1475 (1226–2019)	
>5 per 10 hpf	35	211 (55–367)		304 (371–655)	
Mitotic count			**0.007**		**<0.001**
<1 per 10 hpf	17	NR		3361 (1697–5024)	
1–7 per 10 hpf	45	NR		1019 (639–1399)	
>7 per 10 hpf	28	208 (57–359)		297 (263–331)	
LN metastasis			**0.03**		0.35
No	34	NR		1064 (702–1425)	
Yes	30	173 (89–256)		358 (0–757)	
Radiation therapy			0.28		0.62
No	79	166 (0–356)		601 (190–1012)	
Yes	11	1070 (NR)		953 (640–1266)	
Neoadjuvant VBL			0.16		0.29
No	72	NR		984 (677–1291)	
Yes	18	292 (102–482)		528 (138–918)	

Abbreviations: CI = confidence interval; histo = histologic; hpf = high-powered fields; LN = lymph node; NR = not reached; OST = overall survival time; PFI = progression free interval; VBL = vinblastine; Bold results = statistical significance.

**Table 5 vetsci-11-00363-t005:** Results of multivariable analysis of potential prognostic factors for progression-free interval and overall survival in 90 dogs with cMCT treated with surgery and vinblastine.

Variable	Hazards Ratio	95% Confidence Interval	*p*-Value	Hazards Ratio	95% Confidence Interval	*p*-Value
	Progression-Free Interval			Survival Time		
Model #1						
Age	1.1	0.9–1.2	0.27	1.1	1.0–1.3	0.06
LN positive	3.5	1.5–8.1	**0.003**	1.6	0.8–3.2	0.16
Incomplete local control	2.0	0.9–4.6	0.11	1.5	0.7–3.0	0.26
Kiupel high grade	2.5	1.0–5.8	**0.04**	3.2	1.4–7.2	**0.004**
Model #2						
Age	1.2	1.0–1.3	0.40	1.2	1.1–1.4	**0.001**
LN positive	2.5	1.2–5.2	**0.01**	1.7	0.9–3.0	0.90
Incomplete local control	1.2	0.6–2.5	0.60	1.2	0.6–2.2	0.57
Patnaik grade 3	2.4	1.1–5.0	**0.02**	2.9	1.5–5.7	**0.002**
Model #3						
Age	1.1	1.0–1.3	**0.03**	1.2	1.1–1.4	**<0.001**
LN positive	2.5	1.2–5.2	**0.02**	1.4	0.8–2.6	0.23
Incomplete local control	1.1	0.5–2.4	0.74	1.1	0.6–2.1	0.76
Mitotic count > 5	3.0	1.4–6.3	**0.003**	2.8	1.4–5.5	**0.003**
Model #4						
Age	1.2	1.0–1.3	**0.02**	1.3	1.1–1.4	**<0.001**
LN positive	2.8	1.2–6.1	**0.009**	1.6	0.9–2.8	0.14
Incomplete local control	1.0	0.5–2.1	0.95	1.0	0.5–1.9	0.97
Mitotic Count > 7/10 hpf	7.1	2.1–23.8	**0.001**	9.1	3.0–27.0	**<0.001**
1–7/10 hpf	2.1	0.7–6.5	0.19	2.9	1.1–7.6	**0.03**
<1/10 hpf	1.0			1.0		

Abbreviations: hpf = high powered field; Bold results = statistical significance, # = number.

## Data Availability

The original contributions presented in the study are included in the article, further inquiries can be directed to the corresponding author.

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
