# Peer review of "Tumor Grade and Mitotic Count Are Prognostic for Dogs with Cutaneous Mast Cell Tumors Treated with Surgery and Adjuvant or Neoadjuvant Vinblastine Chemotherapy"

_vetsci, 2024, doi:10.3390/vetsci11080363_

Round 1

Reviewer 1 Report

Comments and Suggestions for Authors

First of all, I would like to thank you for inviting me to review the manuscript entitled “Tumor Grade and Mitotic Count are Prognostic for Dogs with Cutaneous Mast Cell Tumors Treated With Surgery and Adjuvant or Neoadjuvant Vinblastine Chemotherapy.”

The manuscript concerns a study of canine cutaneous mast cell tumors that underwent surgery and neoadjuvant or adjuvant vinblastine chemotherapy. Different prognostic factors that influenced progression free interval and overall survival time were evaluated. The manuscript is very well written. Despite the fact that there are some limitations to this study it adds significant information compared to the other published material. For example the grading systems used (Kiupel and Patnaik) are prognostic for survival. On the other hand neoadjuvant chemotherapy does not influence survival. The conclusions are consistent with the evidence and arguments presented. The references are appropriate.

I can find no major flaws in the manuscript and I suggest that it is accepted without alterations.

Author Response

On behalf of the authors, we would like to thank this reviewer for their time and comments!

Reviewer 2 Report

Comments and Suggestions for Authors

Dear Authors,

Thank you for submitting your manuscript to Veterinary Sciences.

I have completed my evaluation of your manuscript and, though I recognize the value of your work, I’m going to recommend publication upon major revision.

I strongly encourage the authors to revise the manuscript, considering all issues mentioned in my comments carefully, and to resubmit it after doing so. Please outline every change made in response to my comments and provide suitable rebuttals for any comments not addressed.

Kind regards,

The Reviewer

Comments on the Quality of English Language

Small typos should be corrected.

Author Response

Please see the document below.  The authors would like to thank the reviewer for their very thorough review and time.

Reviewer 3 Report

Comments and Suggestions for Authors

Author Response

Thank you to the reviewer for their comments.  Our response is below:

Lines 99-100: If one of the objectives was to evaluate the prognostic factors that influenced progression free interval (PFI) and overall survival time (OST) in a contemporary population of 90 dogs with cMCT that underwent surgery and received vinblastine, why accept dogs that could also receive radiotherapy? How to distinguish the effect of one from the other?

Radiation therapy was included when used to achieve adequate local control (ALC) consistent with previous studies evaluating vinblastine and prednisone in cMCTs (Thamm et al., JVIM 1999 and J Vet Med Sci 2006; Vickery et al., VCO 2006). We have clarified this within the Materials & Methods section. Further details about the radiation therapy that patients received were added to the manuscript; no dogs received RT in the neoadjuvant setting and all dogs that got neoadjuvant VBL did not have RT until after their surgery.  The impact of RT on survival was calculated and added to Table 4.

Line 256: ...and with Patnaik grading?

The beginning of the sentence (lines 254-255) refer to Patnaik grading

Reviewer 4 Report

Comments and Suggestions for Authors

n this retrospective study, the authors aim to evaluate the prognostic factors affecting the outcomes of canine cutaneous mast cell tumors treated with surgery or vinblastine. The manuscript is clearly written and provides a thorough analysis of the data. However, the primary reason for recommending rejection is the absence of novel findings. The study's results align with previously reported data, and it is well-established that histological grading and mast cell count are strong prognostic factors. Additionally, the efficacy of vinblastine in reducing tumor size is already known, and as the authors mentioned, the small sample size of vinblastine-treated cases makes it difficult to draw definitive conclusions. The interpretations offered do not significantly advance the current understanding or provide new insights regarding prognostic factors. Thank you for the opportunity to review this manuscript, and I look forward to seeing future work from the authors.

Author Response

In this retrospective study, the authors aim to evaluate the prognostic factors affecting the outcomes of canine cutaneous mast cell tumors treated with surgery or vinblastine. The manuscript is clearly written and provides a thorough analysis of the data. However, the primary reason for recommending rejection is the absence of novel findings. The study's results align with previously reported data, and it is well-established that histological grading and mast cell count are strong prognostic factors. Additionally, the efficacy of vinblastine in reducing tumor size is already known, and as the authors mentioned, the small sample size of vinblastine-treated cases makes it difficult to draw definitive conclusions. The interpretations offered do not significantly advance the current understanding or provide new insights regarding prognostic factors. Thank you for the opportunity to review this manuscript, and I look forward to seeing future work from the authors.

The authors would like to thank the reviewer for their time and honesty.  In our eyes, the purpose of the paper was to re-evaluate some of the findings from papers that are now 20+ years old and with updated histopathology parameters such as mitotic count which were not recorded in the initial populations.

Round 2

Reviewer 2 Report

Comments and Suggestions for Authors

Dear Authors:

Thank you for your great effort in responding to the previous revision.

I believe your paper may be published after some revision.

Comments on the Quality of English Language

Author Response

Dear Authors: 

Thank you for your great effort in responding to the previous revision. I believe your paper may be published after some revision. 

Thank you again for your time in reviewing our manuscript.  Point responses are included below.

Along the text uniformize how you write numbers and units (should be separated, but at least should be uniformized). Ex.: Line 278-280 (range 1.6-2.7mg/m2). Dose escalation was at the discretion of the clinician and the maximally tolerated dose ranged from 1.4-3.8 mg/m2. 

We have looked through the document and corrected these.  

Line 125 not all dogs ( ) successfully sample) . This sentence should be passed to Discussion (maintain only what you did and not the explanations why you did it this way; these are discussion). 

The purpose of these statements in Materials & Methods was to explain the philosophy behind staging; while it was up to the discretion of the overseeing clinician, these general principles are followed by clinicians within our institution. 

Line 126 spleen or liver aspirates to assess metastatic disease were not routinely performed if there were no ultrasonographic concerns. This statement goes against the one below where the authors say that 50% of the dogs (n=41) had ultrasonographic changes [line 224]. 

The 50% of dogs that had ultrasonographic changes included mild changes in non-target organs (such as renal changes) or mild lymphadenopathy that was commented on but unable to be sampled due to small size.  The wording for this has been changed to attempt clarification both in lines 126 as well as 223-225.  

Line 220 one dog (5%) had cytologic confirmation of metastasis ( ) 41 (50%) were considered normal, while mild heterogenous hepatopathies, enlarged medial iliac lymph nodes, splenic nodules, and mild degenerative renal disease were noted in the remaining dogs. As far as I understand, this means that 50% of the dogs (n=41) had ultrasonographic changes. So, why do you say that aspirates are obtained based upon imaging findings. ? [line 550] Although I recognize the value of the study, when you present a study about the relation between some prognostic factors and the evolution of cMCT in dogs, where OST and PFI are used as variables, the fact that distant metastases are underdiagnosed, and this is not deeply assessed, is a vulnerability. This must be further explored in discussion. 

Wording of this section has been clarified/expanded and further discussion has been added (lines 550-560).  

Line 236 the relation between the histopathologic grade and distant metastasis, in my opinion, should be withdraw since these animals weren t well investigated for such. Consider adding we sought ( ) and local metastasis. 

Line 249 stats for distant metastasis should be withdraw for the same reason abovementioned. 

While evaluating metastases is not the main purpose of our study, we were still interested in this piece of information, as we expect our readership would be.  We have reworded this and added additional discussion of limitations but elect to keep these lines within the manuscript. 

Line 550 At the University of Wisconsin Veterinary Care Hospital, liver and spleen aspirates are not performed in every case and instead aspirates are obtained based upon imaging findings. . But above [line 224] you stated that there were ultrasonographic changes in 41 animals. 

Similar comment to that for line 220.  Additional discussion of limitations has been added. 

Line 588: Informed Consent Statement I am not aware of the US legislation regarding this matter, but in Europe there are strict rules and laws when using information from patients and owners. This usually is secured by an authorization that owners give institutions allowing the use of data for some specified uses. 

In the United States, as long as no identifying features are published (i.e. name or photos), patient data can be used for retrospective studies without informed consent. 

Line 315 Table 3 row 4 Not assessed 5 dogs (%?), 1(8%); 0(0%) 

The table has been updated. 

Line 381 Table 4 - row 3 Kiupel instead of Kuipel  

This has been changed. 

Line 538 OST instead of MST 

This has been changed 

Line 654 Reference 29 – review 

This has been altered. 

Reviewer 4 Report

Comments and Suggestions for Authors

The authors emphasize the retrospective nature of their study, re-evaluating a prognostic factor on a topic over 20 years old. However, the manuscript does not significantly advance the field or provide novel insights beyond established knowledge. Therefore, I maintain my recommendation for rejection as the study lacks novelty and does not contribute sufficiently to the journal’s scope of publishing novel findings or significant reinterpretations of existing data.

Author Response

The authors emphasize the retrospective nature of their study, re-evaluating a prognostic factor on a topic over 20 years old. However, the manuscript does not significantly advance the field or provide novel insights beyond established knowledge. Therefore, I maintain my recommendation for rejection as the study lacks novelty and does not contribute sufficiently to the journal’s scope of publishing novel findings or significant reinterpretations of existing data.

The authors respectfully disagree with reviewer 4 and believe that there have been significant changes in the field of veterinary medicine, including surgical oncology technique, management of chemotherapy patients, and owner decision making in regards to choosing more aggressive therapy for their pets, and therefore we feel a fresh look at prognostic factors in this specific population of patients was warranted and helpful for practicing clinicians.  The authors continue to be thankful for the time the reviewer spent on evaluating our manuscript, and recognize that respectful disagreement and discussion of varying view points is what promotes a healthy scientific community.

Round 3

Reviewer 2 Report

Comments and Suggestions for Authors

Dear Authors: 

Thank you for your great effort in responding to the previous revision. I believe your paper may be published.

Kind regards,

The Reviewer. 

Author Response

Thank you again for your time and effort in improving our manuscript.